# Feasibility of Poly (Vinyl Alcohol)/Poly (Diallyldimethylammonium Chloride) Polymeric Network Hydrogel as Draw Solute for Forward Osmosis Process

**DOI:** 10.3390/membranes12111097

**Published:** 2022-11-03

**Authors:** Ananya Bardhan, Senthilmurugan Subbiah, Kaustubha Mohanty, Ibrar Ibrar, Ali Altaee

**Affiliations:** 1Department of Chemical Engineering, Indian Institute of Technology Guwahati, Guwahati 781039, India; 2Centre for Green Technology, School of Civil and Environmental Engineering, University of Technology Sydney, 15 Broadway, Sydney, NSW 2007, Australia

**Keywords:** forward osmosis, draw solute, hydrogel, liquid-food concentrate, SWRO-FO hybrid process

## Abstract

Forward osmosis (FO) has been identified as an emerging technology for the concentration and crystallization of aqueous solutions at low temperatures. However, the application of the FO process has been limited due to the unavailability of a suitable draw solute. An ideal draw solute should be able to generate high osmotic pressure and must be easily regenerated with less reverse solute flux (RSF). Recently, hydrogels have attracted attention as a draw solution due to their high capacity to absorb water and low RSF. This study explores a poly (vinyl alcohol)/poly (diallyldimethylammonium chloride) (PVA-polyDADMAC) polymeric network hydrogel as a draw solute in forward osmosis. A low-pressure reverse osmosis (RO) membrane was used in the FO process to study the performance of the hydrogel prepared in this study as a draw solution. The robust and straightforward gel synthesis method provides an extensive-scale application. The results indicate that incorporating cationic polyelectrolyte poly (diallyldimethylammonium chloride) into the polymeric network increases swelling capacity and osmotic pressure, thereby resulting in an average water flux of the PVA-polyDADMAC hydrogel (0.97 L m^−2^ h^−1^) that was 7.47 times higher than the PVA hydrogel during a 6 h FO process against a 5000 mg L^−1^ NaCl solution (as a feed solution). The effect of polymer and cross-linker composition on swelling capacity was studied to optimize the synthesized hydrogel composition. At 50 °C, the hydrogel releases nearly >70% of the water absorbed during the FO process at room temperatures, and water flux can be recovered by up to 86.6% of the initial flux after 12 hydrogel (draw solute) regenerations. Furthermore, this study suggests that incorporating cationic polyelectrolytes into the polymeric network enhances FO performances and lowers the actual energy requirements for (draw solute) regeneration. This study represents a significant step toward the commercial implementation of a hydrogel-driven FO system for the concentration of liquid-food extract.

## 1. Introduction

Forward osmosis (FO) is a membrane process in which the concentrated draw solution (DS) and dilute feed solution (FS) are separated by a semi-permeable membrane, capable of allowing the passage of water while rejecting salts. The spontaneous movement of water molecules from FS to DS is driven by the osmotic pressure gradient between the two solutions, as opposed to other pressure-driven membrane processes, such as microfiltration, ultrafiltration, nanofiltration, and reverse osmosis. The FO process offers low fouling propensity, high salt rejection, and low brine discharge [1]. Reportedly, compared to conventional concentration techniques, the FO process is expected to be energy-efficient, provided that draw solution (DS) regeneration is achieved at a low cost, i.e., either by using waste heat resources or thermal and electrical stimulus (which requires less energy than thermal evaporation methods) [2]. Recently, the FO process has received significant attention from researchers and has been reported to have potential applications in wastewater treatment [3], desalination [4], food processing [5], energy production [6], and biomedical applications [7]. 

Despite intense research, commercial and industrial applications of the FO process have been restricted due to certain obstacles, including the separation and regeneration of draw solute, concertation polarization, membrane fouling, and, most importantly, reverse solute flux (RSF) [8]. The significant parameter constraining the commercialization of the FO process includes the unavailability of a suitable draw solute and an efficient FO membrane [9]. In the FO process, osmotic pressure is the primary driving force for mass transport. The type and concentration of draw solute play a significant role in affecting overall FO performance. 

An ideal draw solute should generate high osmotic pressure, low RSF, minimal toxicity, and be cheap. Reportedly, inorganic salts [10,11,12], such as NaCl, MgCl_2_, CaCl_2_, KBr, MgSO_4_, and (NH_4_)_2_CO_3_, are the most extensively investigated DSs for FO applications. In recent years, numerous synthetic draw solutes (polyelectrolytes, magnetic nanoparticles, switchable polarity solvents, ionic liquids, and polymeric hydrogels) have been developed and investigated in the FO process [13,14,15,16,17]. High osmotic (or swelling) pressure, negligible reverse solute flux, and draw solute recovery are a few potential advantages of using a polymer hydrogel as a draw solute [8]. Li et al. (2019) [18] reviewed recent developments and improvements in FO membrane technology. Polymeric hydrogels are conventionally described as a three-dimensional interconnected polymeric network, capable of swelling and retaining a significant amount of water in its structure [19]. In hydrogels, the capacity to entrap a large volume of water is caused by the hydrophilicity and flexibility of the polymeric network [20]. The polymer hydrogel’s presence and dissociation of ionic species induce the hydrogel swelling and the development of internal osmotic pressure [21]. Due to the unique structural characteristics and the overall FO performance, the porous hydrogel has recently developed as a research interest. In any polymeric hydrogel, water absorption largely depends on its swelling (osmotic) pressure, which is the key driving force of any FO process. 

Compared to conventionally used draw solutes in the FO process, the hydrogels are not in a liquid state and have considerably higher viscosity. The role of hydrogels as draw solutes in the FO process has been explained in a recent review article by Wang et al. (2020) [9]. 

Hydrogels with a high swelling pressure drive water molecules from the FS through a selective membrane while the salts are rejected. The swelling behavior of hydrogels strongly depends on the cross-linking density and interaction between the solvent molecules and polymeric chain segments. The cross-linking density plays a crucial role in the swelling properties of the hydrogel [22]. It is reported that, with a low cross-linking density, water molecules cannot be held inside the polymeric network, whilst a highly cross-linked polymeric network does not allow the entry of water molecules.

The ionic polymer hydrogel with a thermal responsive unit was first investigated by Li et al. (2011) [23]. Reportedly, the combination of pressure and thermal stimuli induced the higher water permeation and higher dewatering capability of the polymeric network. Cai et al. (2013) [13] synthesized thermally responsive semi-IPN hydrogels synthesized by the polymerization of N-isopropyl acrylamide (NIPAm) in the presence of poly sodium acrylate (PSA) or poly (vinyl alcohol) (PVA). Zhang et al. (2015) [24] synthesized a series of electrically responsive hyaluronic acid–polyvinyl alcohol (HA/PVA) polymeric hydrogels, prepared using repetitive freezing and thawing cycles, which were used as the DSs for the FO process. The reported initial water flux was lower than 2 L m^−2^ h^−1^. 

Polyvinyl alcohol (PVA) hydrogels are widely used in pharmaceutical and biomedical applications. Cross-linking reactions are one of the most commonly used techniques to improve the physical properties of PVA. Hosseinzadeh (2013) [25] described the preparation and characterization of a superabsorbent hydrogel made of PVA cross-linked with glutaraldehyde (GLA). Reportedly, when dialdehydes (such as glutaraldehyde and glyoxal) are used, the cross-linking reaction of PVA can be conducted under mild conditions [26]. Poly (diallyldimethylammonium chloride) (PolyDADMAC), a cationic polyelectrolyte, was the first polymer to be permitted for use in potable water treatments by the Food and Drug Administration (FDA), USA. Reportedly, a strong cation electrolyte (such as a quaternary ammonium group, C_8_H_6_N^+^) results in a DS with high electrical conductivity. Hamad and Chirwa (2019) [27] investigated the performance of the cationic polyelectrolyte PolyDADMAC as an osmotic agent for the FO process. 

This study reports the preparation and characterization of a PVA-PolyDADMAC-based hydrogel. The effect of polymer concentration, cross-linker concentration, and FS concentration was observed to analyze the hydrogel performance as a DS in the FO process. The FO process was carried out at room temperature, using a Filmtech RO membrane for two FS concentrations of 2500 mg L^−1^ and 5000 mg L^−1^ NaCl. The synthesized hydrogel’s swelling capacity and life-cycle assessment were also investigated to explore its practical applicability. 

## 2. Material and Methods

### 2.1. Materials

PVA (average M_w_ 13,000–23,000, 98% hydrolyzed) and PolyDADMAC (average M_w_ 200,000–350,000) were purchased from Sigma-Aldrich (St. Louis, MO, USA) to synthesize the hydrogels. Glyoxal (40% aqueous solution, cross-linker), a cross-linking agent, was purchased from SRL chemicals. Sodium chloride purchased from Merck (Darmstadt, Germany) was used to prepare the FS. 

### 2.2. Synthesis of Hydrogel

A series of hydrogels were synthesized to understand the effect of polymer (PVA and PolyDADMAC) and cross-linker (Glyoxal) concentration on the overall swelling capacity of the hydrogels (Table 1). 

The polymers were allowed to form an aqueous solution with 30 mL of deionized (DI) water at 70 °C for 3 h. The cross-linker was added to the PVA-PolyDADMAC solution to maintain the same operating conditions. After 45 min, the resultant polymeric solution was allowed to cool down to room temperature, and then it was poured into non-solvent ethanol (500 mL) for 24 h to remove the absorbed water. Then, the ethanol was decanted, and the product was cut into small, uniform-sized pieces and dried by blowing warm air at 30 ± 1.5 °C for 8 h. After drying, the hydrogel was placed in a desiccator until further use (Figure 1). 

### 2.3. Characterization of Hydrogel

The surface morphology of the prepared hydrogel was characterized using a field-emission scanning electron microscope (FESEM, Make: Zeiss, Jena, Germany; Model: Sigma). The images were obtained at an accelerated voltage of 3 kV at different magnifications. The amorphous and crystalline structures of the prepared hydrogel were examined using XRD. After each cycle, the crystalline nature of the gel was analyzed using an X-ray diffraction (XRD; make: Bruker, Billerica, MA, USA; model, D8 Advance) pattern. Further, FTIR (Model No.: IRAffinity-1; Make: M/s Shimadzu, Kyoto, Japan) analysis was used to characterize the presence of specific chemical groups in the synthesized hydrogel. 

### 2.4. The Swelling Ratio of Hydrogel

The swelling ratio of the prepared hydrogel was measured against DI. A total of 2 g of hydrogel was placed in a dialysis bag and immersed in a beaker of 500 mL of DI water or salt solution for 3 h at room temperature to determine the hydrogel’s swelling ratio (Q, g g^−1^). The swelling ratio of the given hydrogel was calculated as follows:(1)$$Q=\frac{\left(W_{S}-W_{D}\right)}{W_{D}}$$
where *W_S_* and *W_D_* are the weight of the swollen and dried hydrogel (in g).

The swelling ratio (Q, g g^−1^) can be defined as the fractional increase in the weight of the hydrogel due to water absorption. A specified amount of swollen hydrogel was placed under a blower at room temperature for the dewatering process. The hydrogel’s dewatering continued until a weight change between two readings was constant. 

### 2.5. FO Performance

In the FO process, the membranes can either be used with an active layer (AL) facing the FS (AL-FS or FO-mode) or with an AL facing the DS (AL-DS or PRO-mode) (Figure 2) to determine the effect of membrane orientation on overall FO performance. At room temperature, a certain amount of dry hydrogel (0.5 g) was mixed with 0.5 mL of deionized water. Depending on the membrane orientation, the semi-swollen hydrogel was placed uniformly on the membrane surface using a flat-surfaced spatula. Reportedly, in AL-DS mode, the dilutive external concentration polarization is less severe and, as a result, the permeate flux is higher in AL-DS (PRO) mode than in AL-FS (FO) mode [28,29]. However, AL-FS mode is widely used because fouling can be removed easier from the dense AL than from the porous support layer (SL) [30]. In the FO mode of the operation, the concentration polarization (CP) effect is expected to reduce near the semi-swollen hydrogel (DS) at the membrane surface, which enables high ΔΠ_PRO-Mode_, rather than ΔΠ_FO-Mode_ [31,32]. Reportedly, due to high molecular weight and viscosity, the diffusivity of the hydrogel is expected to be lower than NaCl because ICP is the major factor contributing to the decline in the water permeation rate in FO desalination [9]. Thus, considering the severity of the ICP effect, the AL-DS mode of the FO operation was chosen for further experimental studies. 

Figure 3a,b provides a pictorial representation of the synthesized polymeric network before and after drying. The FO tests were carried out in a thimble-like FO membrane module (Figure 3c). The thimble-like membrane was prepared by cutting the membrane (length: 6 cm; breadth: 3.5 cm) and gluing it together to form a pouch-like structure. After gluing, the active dimension of the membrane was measured accurately by excluding the glued portion (area: 17.5 cm^2^). Before experiments, a study was performed to detect leakage in the prepared membrane module. The module without leakage was stabilized for another 8 h using deionized distilled water (TDS: 0.445 mg L^−1^).

Figure 3d provides a schematic representation of the batch-scale FO setup. A specific amount of hydrogel (in a semi-swollen state) (Figure 3b) was used as a draw solute (DS) facing the AL of the membrane surface (AL-DS mode) to determine FO performance. An aqueous solution of a NaCl solution at room temperature was used for the FS. Due to the osmotic pressure gradient between the FS and the hydrogel, the solvent from the FS flowed across the other side of the membrane to the hydrogel (draw solute). In our study, the membrane allowed the selective permeation of water (solvent) from the FS to permeate through the membrane, while rejecting the hydrated ions from the FS and hydrogel. The same batch of FO experiments was performed using a high-concentration NaCl solution as the DS and deionized water as the FS to investigate the role of the hydrogel as a draw solute. 

The change in concentration of the sodium-ion and chloride-ion in the FS was determined using ion chromatography (M/s Metrohm, Herisau, Switzerland) and a benchtop conductivity meter (M/s HANNA Instruments, Woonsoket, RI, USA). 

Osmotic pressure is the minimum pressure that must be applied to a solution to prevent the solvent from flowing through the semi-permeable membrane.
(2)Osmotic pressure, π=n×ϕ×c×R×T
where *n* = dissociation factor; *Φ* = osmotic coefficient of the solute, *c* = molar concentration of the solution (mol L^−1^), *R* = universal gas constant (8.314 J K^−1 ^mol^−1^), and *T* = temperature (K). 

The osmotic pressure of the FS in this study was determined using a freezing point osmometer (Osmometer basics, Type 7/7/M/s Löser Messentechnik, Berlin, Germany). The test cell, consisting of the draw solute, was placed in a closed chamber filled with FS, and the setup was adequately sealed with paraffin to avoid evaporation loss. The permeate flux (L m^−2^ h ^−1^), was determined by the change in the weight of the hydrogel over a certain period.
(3)$$\mathrm{Permeate}\;\mathrm{flux},\;J_{w}=\frac{\Delta W}{\Delta t\times A_{m}\times \rho }\left[\frac{L}{m^{2}\cdot h}\right]$$
where Δ*W* (kg) is the weight change of the test cell due to the permeation of water through the FO membrane over a pre-determined period (Δ*t*, h), *A_m_* (m^2^) is the effective membrane area of the FO test cell, and *ρ* is the density of the FS (kg m^−3^).

For the estimation of RSF, instead of a NaCl solution, DI water was used as the FS. The reverse diffusion of the DS component was observed by measuring the total dissolved solids (TDS, mg L^−1^) in the FS tank. The TDS was measured using a digital conductivity meter (HANNA edge^®^ M/s HANNA Instruments, India).
(4)$$\mathrm{Reverse}\;\mathrm{solute}\;\mathrm{flux},\;J_{s}=\frac{\Delta (C\times V)}{\Delta t\times A_{m}}\left[\frac{g}{m^{2}\cdot h}\right]$$
where Δ*C* (g L^−1^) and Δ*V* (L) are the change in concentration and volume, respectively, of the FS over a pre-determined period (Δ*t*, h) and *A_m_* (m^2^) is the effective membrane area of the FO test cell.

### 2.6. Hydrogel Stability Test

To determine the prepared hydrogel’s practical applicability, the determination of hydrogel reusability is very critical. The reusability study was performed using the method mentioned by Pan et al. (2021) [33] with slight modifications. The performance of the prepared hydrogel was determined in terms of permeate flux against a 5000 mg L^−1^ NaCl solution as the FS. After the FO process, the saturated hydrogels were regenerated by blowing hot air (39–50 °C). The change in weight of the swollen hydrogel was regularly observed, and the process of dewatering was continued until the weight of the hydrogel reached 0.5 ± 0.05 g. After each swelling and deswelling cycle, an FO experiment was performed to verify the prepared hydrogel’s stability. The above FO and regeneration operation were repeated until the average flux became constant.

The reuse ratio is defined as the proportion of average water flux of the *n*^th^-time ($J_{w,n^{th}}$) to the initial cycle (0th cycle, $J_{w,0^{th}}$), and is calculated as follows:(5)$$reuse\;ratio=\left(\frac{J_{w,n^{th}}}{J_{w,0^{th}}}\right)\times 100\%$$

## 3. Result and Discussion

### 3.1. Characterization

#### 3.1.1. Membrane’s Characterization

The membrane used in this study, DOW Filmtech, was supplied by Eureka Forbes (India). The membrane’s filtration properties (in terms of pure water permeability coefficient, solute permeability coefficient, and structural properties) are characterized and reported in Table 2. 

Before any study, the membrane was stabilized at a high pressure. Each experiment was repeated three times for accuracy, and the average values were reported. 

In this study, the basic objective was to evaluate the feasibility of the synthesized hydrogel as a draw solute for the DOW Filmtech membrane. Reportedly, although the given membranes are manufactured for RO applications, they are also expected to provide a decent flux in FO mode. For example, pure water permeability and solute permeability of the commercial flat-sheet cellulose-triacetate (CTA) FO membrane (Model FTS H2O, USA) was 0.52 L m^−2^ h^−1^ bar^−1^, and 0.234 L m^−2^ h^−1^, respectively [34]. The FO performance (in terms of water flux, L m^−2^ h^−1^) for the DOW Filmtech membrane (2.131 L m^−2^ h^−1^) and FTS H2O (2.774 L m^−2^ h^−1^) membrane was found to be almost similar when DI water was used against a 2000 ppm NaCl solution as the draw solution. 

#### 3.1.2. Morphological Characterization of the Synthesized Hydrogel

The morphology of the synthesized hydrogel was observed using field-emission scanning electron microscopy (FESEM) at 500 and 1000× magnification (Figure 4). At the 0th cycle (before the experiment), the FESEM image exhibited an uneven surface with a porous structure on the cross-sectional surface of the synthesized hydrogel. After the 12th cycle of FO and dewatering, the performance of the synthesized hydrogel was reduced. The same was represented using the comparatively smooth surface of the gel with a reduced porous structure on the cross-sectional surface of the hydrogel. The reduced membrane performance was due to the surface morphology of the synthesized hydrogel as a result of the repetitive cycles of swelling and deswelling. The change in the surface morphology of the hydrogel subsequently reduced water flux in the FO process. The reduction in FO performance could also be from dilutive external concentration polarization (DECP) due to the reduced swelling capacity of the hydrogel on the membrane surface. The synthesized hydrogel consists of a loosely cross-linked polymeric network. Due to repeated swelling (in the FO process) and dewatering (for regeneration), the strength of the polymeric chain was expected to break, resulting in a subsequent loss of its porous texture. The FESEM image of the new and used membrane (after 10 cycles) exhibited the deposition of loose hydrogel particles on the membrane surface.

X-ray powder diffraction (XRD) is a rapid analytical technique used for the phase identification of crystalline material. The surface characterization of hydrogel had a significant effect on water absorbency. The change in crystallinity of the hydrogel from cubic (0th–3rd cycle), tetragonal (7th cycle), orthorhombic (9th cycle), and monoclinic (10th cycle) changes can be considered primarily responsible for the reduced water absorption capacity of the synthesized hydrogel.

The ATR-FTIR spectra for the 0th/5th/7th/9th/10th cycles are shown in Figure 5. ATR-FTIR spectroscopy was used to provide information on the functional groups near the surface of an internal reflection element. Here, a 0% transmittance meant that the sample had absorbed all radiation, whereas a 100% transmittance meant that the sample absorbed the same amount of radiation as the reference. The peak observed between 3000 cm^−1^ to 3500 cm^−1^ was related to the –OH stretching of PVA and the aliphatic secondary amine of PolyDADMAC. According to a study reported by Mwangi et al. (2013) [35], the –OH group of the solvent (in our case, water) can potentially result in a peak between 3382 cm^−1^ and 3332 cm^−1^. 

The increment of transmittance from the 0th to the 10th cycle represents the reduction in the swelling capacity of the hydrogel. The C–O (crystallinity) at 1141 cm^−1^ represents the increased crystallinity in transmittance (%T). The characteristic C–C of the conjugated carbon atoms of the polyelectrolyte (polyDADMAC) at 1635 cm^−1^ was found to be the same with no significant change. This implies that each cycle’s FO performance change was primarily due to a change in surface morphology and not due to the leaching of the component from the given hydrogel [36,37]. 

The FESEM, FTIR, and XRD report suggests that although the composition of the hydrogel was maintained, the altered surface morphology of the hydrogel was responsible for the degrading trend in the average FO flux. 

### 3.2. Swelling Characteristics of the Hydrogel

The swelling properties of the synthesized polymeric hydrogel are among the most critical parameters that need to be considered. An interior osmotic pressure gradient can control the swelling capacity of the hydrogel (this is related to the number of ionic functional groups, cross-linking points, the interaction between the polymer solvent, and hydrophilicity). The hydrogel composition with the highest swelling ratio can eventually result in a high water (permeate) flux. The swelling ratio was determined in 3 h. The concentration of PVA and PolyDADMAC used in the hydrogel formulation significantly impacted the overall FO performance of the prepared hydrogel. Reportedly, keeping the polyDADMAC and cross-linker ‘glyoxal’ composition the same, it was observed that with an increase in PVA concentration from 10 wt.% to 13.33 wt.%, the swelling ratio increased from 13.45 g g^−1^ to 20.68 g g^−1^ (Table 1). 

In hydrogel preparation, a cross-linking agent prevents the dissolution of the hydrophilic polymer chains in an aqueous environment. As the cross-linker concentration increased from 5 wt.% to 10 wt.%, the swelling capacity decreased from 13.45 g g^−1^ to 11.77 g g^−1^. A higher concentration of cross-linking agents resulted in a diminished polymeric network space and a reduced water absorption capacity [22,38]. 

### 3.3. FO Performance of Hydrogel

Figure 6a represents the change in water fluxes of the prepared hydrogel as the draw solute and the 5000 mg L^−1^ NaCl solution as the FS at 26 °C, room temperature. In 360 min, the flux changed from 1.30 L m^−2^ h^−1^ to 0.68 L m^−2^ h^−1^. The concentration of cationic polyelectrolyte polyDADMAC played a significant role in determining the overall performance of the hydrogel. As the FO process proceeded, a decrease in water flux was observed with time. This decreasing trend in the water flux was evidently due to the decreased concentration gradient between the FS to the draw solute due to the swelling of the hydrogel.

Similarly, Figure 6b represents the overall performance of the best-performed draw solute against different NaCl (5000 mg L^−1^ and 2500 mg L^−1^) concentrations and DI water as the FS. Conventionally, the FO process occurs when the osmotic pressure gradient between the feed and draw solution is positive (∆π > 0) at the same hydrostatic pressure. Thus, as the FS concentration increased from DI water to 5000 mg L^−1^ NaCl, the corresponding flux (in first hour) decreased from 1.91 L m^−2^ h^−1^ to 1.30 L m^−2^ h^−1^, respectively. The decreased flux, with time, was attributed to the reduced osmotic pressure gradient across the membrane. 

Along with the permeate flux (L m^−2^ h^−1^), the RSF also plays a significant role in determining the overall feasibility of the synthesized hydrogel. In this case, the NaCl solution is replaced by deionized water. The change in conductivity of the deionized water was used to estimate the RSF (0.11 g m^−2^ h^−1^) using Equation (4). In this work, a comparison of FO performance in terms of initial permeate flux (L m^−2^ h^−1^) and water recovery with other reported hydrogels as the draw solute is presented in Table 3.

Reportedly, however, the permeate flux of the synthesized hydrogel was considerably low. The regeneration performance of the hydrogel (>70%) was far superior to the mentioned hydrogel. 

### 3.4. Hydrogel Stability Test

During the swelling process, the water molecules diffused into the polymeric network and, as a result, the polymer chain started to relax. After each FO process, the hydrogels (osmotic agents) were regenerated by blowing hot air (40 °C). The change in the weight of the hydrogel was accurately measured using a digital weighing balance. The process of dewatering the hydrogel continued until the weight change became constant. After the regeneration of the osmotic agent (hydrogel), the dried osmotic agent was subjected to another cycle of the FO process, followed by DS regeneration. This swelling and deswelling of the synthesized hydrogel continued until the change in permeate flux (J_w_, L m^−2^ h^−1^) became constant. Figure 7a represents the change in average flux with each cycle of FO (swelling) and deswelling (regeneration). Additionally, the application of hot air (40 °C) increased the crystallinity and, as a result, reduced the swelling capacity of the hydrogel. Figure 7b represents each cycle’s change in flux (in the first 60 min). The surface morphology of the hydrogel also changed from a porous to a crystalline structure (Figure 7c).

This study suggests that water flux can be recovered to 86.6% of the initial flux after 12 hydrogel (draw solute) regenerations. As a result, it is suggested that, instead of implementing thermal influence (hot air) for draw solute regeneration, the performance of the given hydrogel can be further improved by implementing a non-thermal regeneration technique. 

### 3.5. Prospects of the Synthesized Hydrogel in the Field of Liquid-Food Concentration Application

This section of the given study primarily intends to propose the prospects of the synthesized hydrogel as a potential DS for the preparation of concentrated liquid-food extract using the FO process. The seawater water desalination technique involves the removal of inorganic salts and impurities from seawater (35 g L^−1^) to produce fresh water. Along with fresh water, this process also results in a highly concentrated brine reject with almost two to three times the TDSs of seawater (i.e., approximately 65 g L^−1^), potentially providing high osmotic pressure to the FO process. Figure 8a provides a schematic representation of the proposed design for a three-tier FO membrane module for liquid-food concentration using a synthesized hydrogel as the DS. The high-concentration RO brine (63,500 mg L^−1^ NaCl solution, π = 50.05 bar) was potentially used as a regenerating fluid to draw water from the polymeric network continuously. The high-concentration RO brine and liquid-food concentrates were separated by the hydrogel and FO membrane. Here, due to the osmotic pressure gradient, the solvent moved from liquid food (tea) to hydrogel to RO brine, resulting in the simultaneous dilution and concentration of the RO brine and liquid food, while restricting reverse solute flux (RSF). The osmotic pressure of the synthesized hydrogel was approximately 6.305 bar. The osmotic pressure of the synthesized hydrogel (DS) can be estimated by its potential to draw a solvent from the FS (NaCl solution, 0.5 g L^−1^ to 10 g L^−1^). The osmotic pressure of the RO reject brine, and liquid tea extract was measured using a freezing-point osmometer (Osmometer basics, Type 7/7/M/s Löser Messentechnik, Berlin, Germany). 

In the proposed three-tier FO module, the high-concentration rejects the RO brine and low-concentration liquid-food (tea) flow in the counter-current mode in the top and bottom tier, respectively. The middle tier consists of a partially swollen hydrogel, separated by the top and bottom tier using an FO membrane. Using the given hydrogel, a 100 g L^−1^ concentration tea extract (TDS: 1.147 g L^−1^, π = 2.00 bar) can be concentrated, only up to 333 g L^−1^ (TDS: 3.81 g L^−1^, π = 5.00 bar). However, a further concentration of the tea extract (up to 556 g L^−1^) could be possible with hydrogel synthesis with improved osmotic pressure (~9.46 bar), thus providing a future scope for the author to develop a hydrogel with improved swelling and osmotic pressure. 

Figure 8c represents the change in osmotic pressure concerning the change in concentrate volume (for 100 g L^−1^) of the tea extract. Compared to using the direct application of RO brine as the DS for liquid-food concentration, the hydrogel ensured lower RSF. Due to the swelling pressure and osmotic pressure gradient, the solvent from the liquid food permeated the hydrogel. A high-concentration RO brine stream (π = 50.05 bar) ensured the continuous permeation of the solvent from the hydrogel network. Here, because the gel was dewatered using an osmotic pressure gradient instead of thermal treatment, the chances of degradation in the overall hydrogel structure were reduced. In another aspect, seawater can also be used as the DS. However, the cost associated with seawater treatments can potentially make the overall FO process quite expensive. However, the effective use of the RO reject brine ensured a minimized FO operational cost because the brine was pretreated, and it can be directly used for the FO process. The waste RO brine stream, without thermal treatment, ensured effective DS (hydrogel) regeneration. Additionally, the high osmotic pressure that generated the RO reject stream provided the essential driving force for continuous processing. Another benefit of using RO brine as the DS is that concentrated brine can be efficiently diluted before the environment, enabling effective brine management by maintaining the ocean ecosystems without a much localized impact.

## 4. Conclusions

This study synthesized a foam-like porous hydrogel using poly (vinyl alcohol)/poly (diallyldimethylammonium chloride) as polymers. The hydrogel used in this process was synthesized using a robust and straightforward method. The components used for hydrogel synthesis were GRAS (generally recommended as safe). 

Due to the cationic polyelectrolyte ‘polyDADMAC’, the swelling ability (13.6 g g^−1^) and FO performance (1.81 L m^−2^ h^−1^, against 2500 mg L^−1^ NaCl solution) were improved.Further, the saturated hydrogel after the FO process can be regenerated (>70%) by blowing hot air (at 39–50 °C).The study suggests that water flux can be recovered by up to 86.6% of its initial value in 12-times regeneration using the same hydrogel. Thus, the synthesized hydrogel has tremendous potential for lowering energy consumption in FO applications.

This study confirmed the synthesis of an ideal DS, fulfilling all three primary criteria for an appropriate food-grade DS with high osmotic pressure, low RSF (0.11 g m^−2^ h^−1^), and an easy/cost-effective recovery. Furthermore, the flux of the synthesized hydrogel needs to be further improved to make the synthesized hydrogel more feasible for commercial applications. 

## Figures and Tables

**Figure 1 membranes-12-01097-f001:**
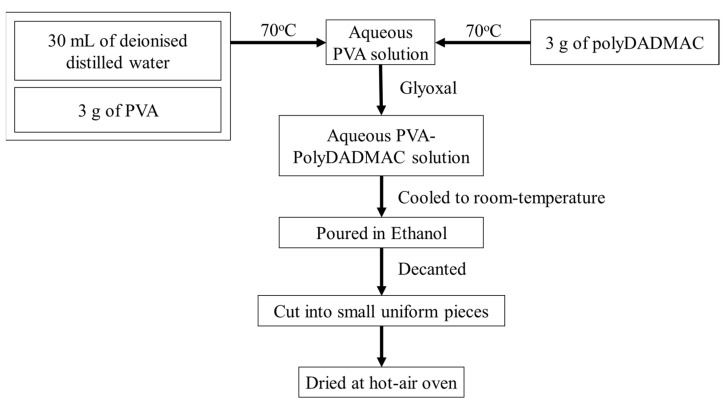
Schematic representation of the method of hydrogel synthesis.

**Figure 2 membranes-12-01097-f002:**
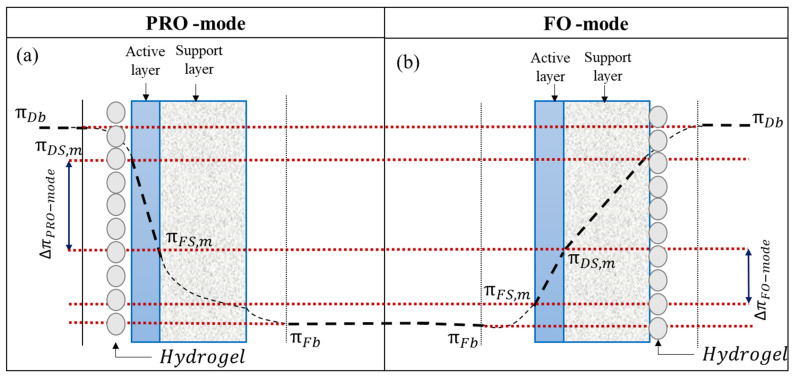
Solute concentration profile for (**a**) AL-DS (PRO)-mode and (**b**) AL-FS (FO)-mode [Note: osmotic pressure gradient, ∆π = ($\pi_{DS,b}$ − $\pi_{FS,b})$ and effective osmotic pressure gradient ∆$\pi_{eff}=(\pi_{DS,m}$ − $\pi_{FS,m})$ ].

**Figure 3 membranes-12-01097-f003:**
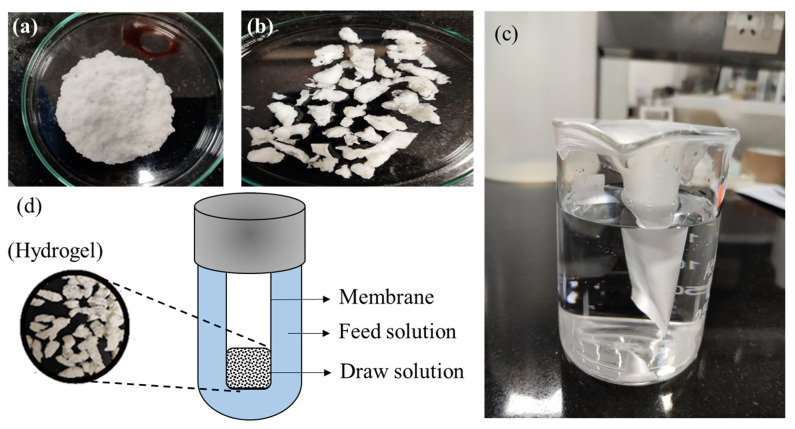
(**a**) Freshly synthesized hydrogel, (**b**) synthesized hydrogel dried at 30 (±1.5) °C for 8 h, (**c**) lab-scale setup, and (**d**) schematic representation of lab-scale setup for forward osmosis using hydrogel as draw solute.

**Figure 4 membranes-12-01097-f004:**
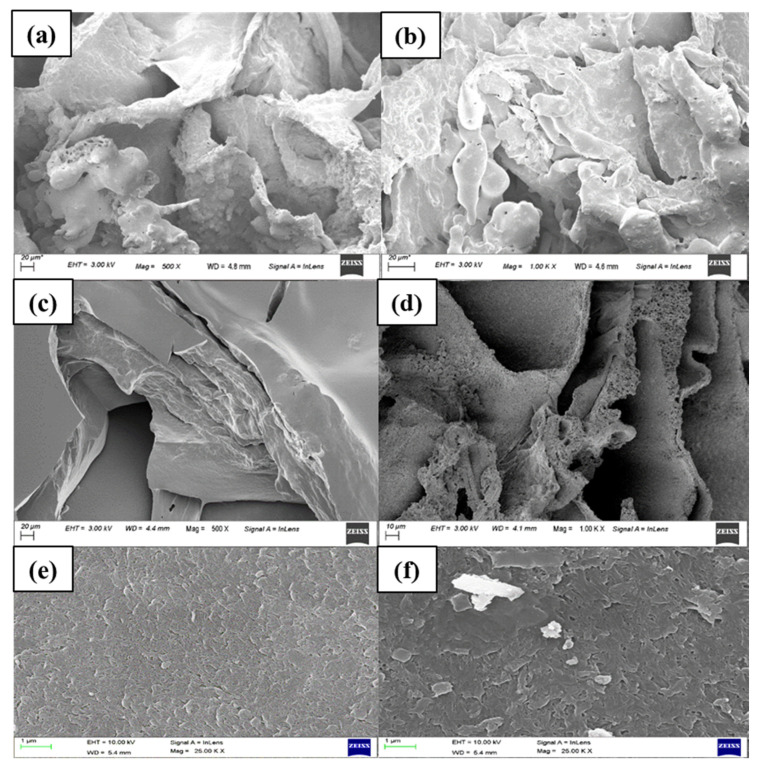
FESEM image of hydrogel before experiment (**a**) at 500×, (**b**) at 1000×, after experiment (12th cycle), (**c**) at 500×, (**d**) at 1000×, image of (**e**) new membrane, (**f**) used membrane.

**Figure 5 membranes-12-01097-f005:**
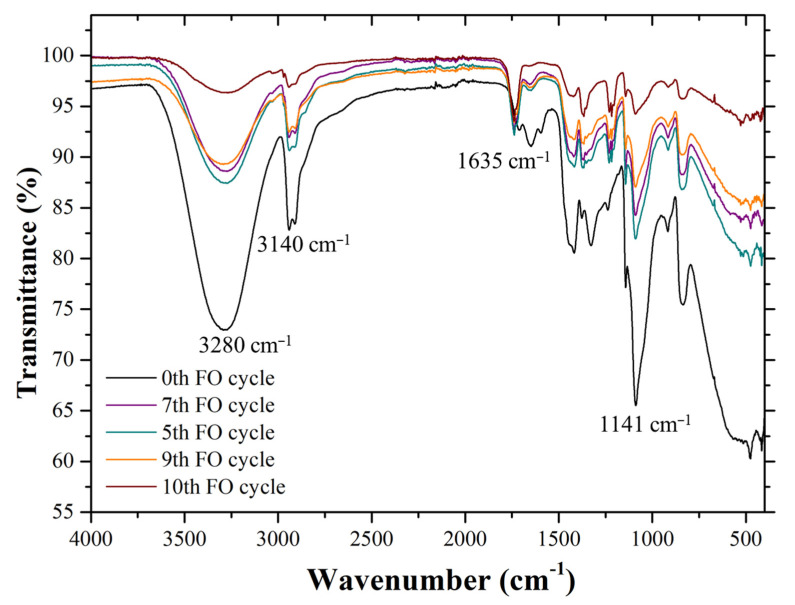
FTIR spectroscopy image of the synthesized hydrogel after each cycle.

**Figure 6 membranes-12-01097-f006:**
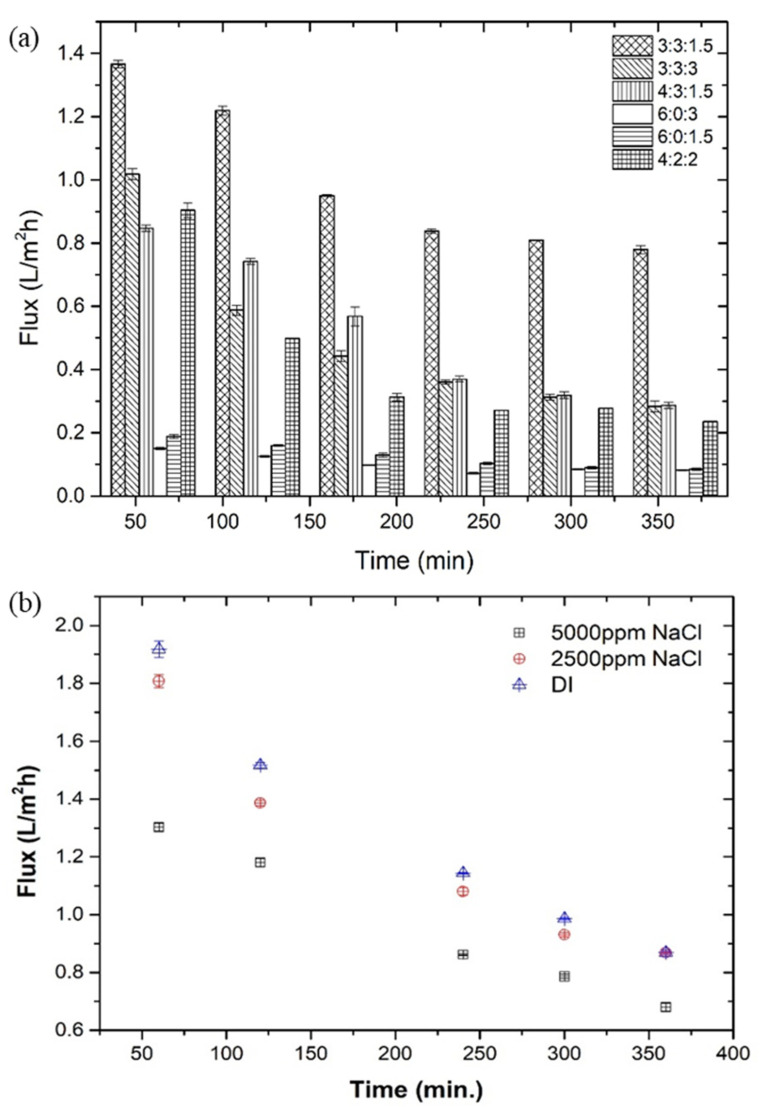
Effect of (**a**) hydrogel composition and (**b**) feed solution concentration on the overall FO performance.

**Figure 7 membranes-12-01097-f007:**
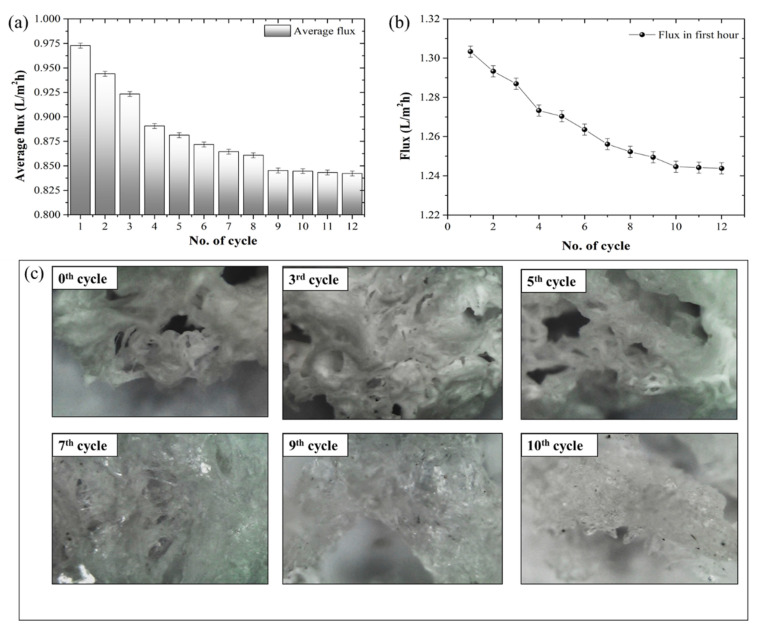
(**a**) Change in average flux with each FO and regeneration cycle, (**b**) change in flux over the first hour, and (**c**) digital microscopic image to determine the change in morphological structure with each cycle, at 100×.

**Figure 8 membranes-12-01097-f008:**
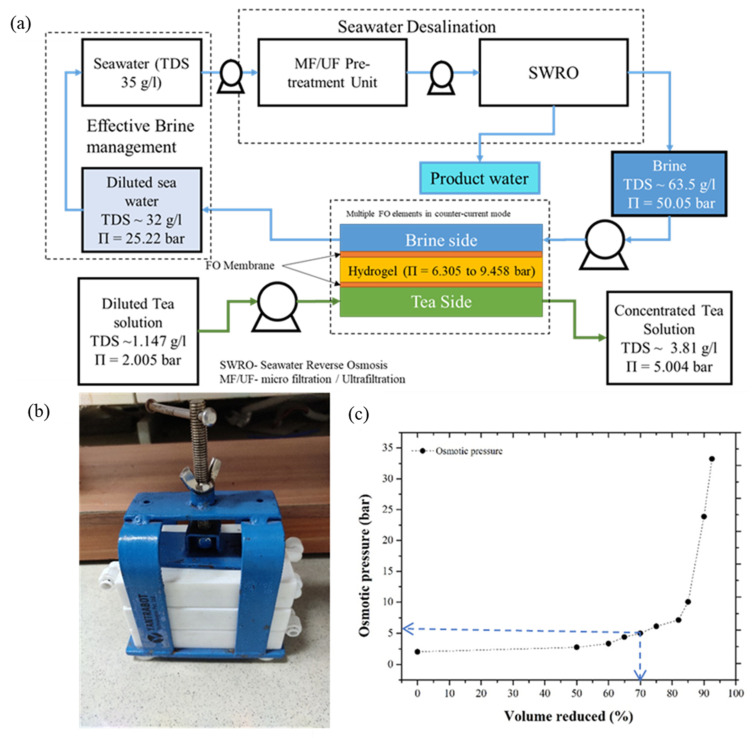
(**a**) Schematic representation of an integrated experimental setup for preparation of concentrated liquid food using hydrogel as DS and RO brine; (**b**) data for correlating change in osmotic pressure with % volume reduced (for tea extract); and (**c**) setup for integrated membrane three-layer FO module for preparation of concentrated liquid food.

**Table 1 membranes-12-01097-t001:** Composition and swelling ratio of the prepared hydrogel.

Code	PVA (wt%)	PolyDADMAC (wt%)	Glyoxal (wt%)	Swelling Capacity (g g^−1^)	Remark
H-3:3:1.5	10	10	5	13.59 ± 0.06	Gel formed with spongy white color texture after drying
H-4:3:1.5	13.33	10	5	20.68 ± 0.06	
H-6:0:1.5	20	0	5	13.45 ± 0.02	
H-4:2:2	13.33	6.67	6.67	16.56 ± 0.05	Gel formed with spongy white color texture after drying
H-3:3:3	10	10	10	11.77 ± 0.02	Gel formed with spongy pale-yellow texture color after drying
H-6:0:3	20	0	10	11.15 ± 0.06	Strong network gel formed

**Table 2 membranes-12-01097-t002:** Properties of the membrane.

Manufacturer	DOW Filmtech Membrane
Material	Polyamide TFC
pH	2–11
Pure water permeability coefficient, A (L m^−2^ h^−1^ bar^−1^)	5.99
Solute permeability coefficient, B (L m^−2^ h^−1^)	3.44
A/B (bar)	0.57
Structural parameter, S (µm)	424–786

**Table 3 membranes-12-01097-t003:** Hydrogel as draw solute for the FO process.

Draw Solute	Feed Solution (mg L^−1^ NaCl)	Regeneration Method	FO Performance		Reference
			Initial Flux (L m^−2^ h^−1^)	Water Recovery (%)	
PSA	2000	Heating at 50 °C	0.96 (in, 60 min)	<5	[23]
HA-PVA-5	2000	Electric field (9V)	1.20	NA	[24]
HA-PVA-7			0.91		
HA-PVA-9			0.90		
SI-0.2PSA	2000	Heating at 40 °C for 10 min	0.18 (in 60 min)	NA	[13]
SI-0.5PSA			0.18 (in 60 min)		
SI-0.2PVA			0.12 (in 60 min)		
PSA/Polyester	2000	Solar simulator (1 kW/m^2^) for 60 min	3.50 (in 60min)	21	[39]
PVA-polyDADMAC	2500	Hot air at 39–50 °C	1.81 (in 60 min)	>70%	(This study)

PSA: poly (sodium acrylate); HA-PVA: hyaluronic acid–poly (vinyl alcohol); SI-0.2PVA: semi-IPN-0.2 polyvinyl alcohol; SI-0.5PVA: semi-IPN-0.5 polyvinyl alcohol; SI-0.2PSA: semi-IPN-0.2 poly (sodium acrylate).

## Data Availability

Not applicable.

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
