# Peer review of "Feasibility of Poly (Vinyl Alcohol)/Poly (Diallyldimethylammonium Chloride) Polymeric Network Hydrogel as Draw Solute for Forward Osmosis Process"

_membranes, 2022, doi:10.3390/membranes12111097_

Round 1

Reviewer 1 Report

Title:

Feasibility of poly (vinyl alcohol)/ poly (diallyldimethylammonium chloride) polymeric network hydrogel as draw solute for forward osmosis process

Ref: membranes-1993034

Overall comments:

The manuscript entitled “Feasibility of poly (vinyl alcohol)/ poly (diallyldimethylammonium chloride) polymeric network hydrogel as draw solute for forward osmosis process is reviewed. Detailed comments are listed in the following section below:

Comments:

1.      Abstract:

-        Forward osmosis (FO) is one of the emerging technologies of what technologies?

-        Why draw solute is important and how it can significantly affect the commercial implementation? What kind of process? Please be specific.

-        contains a sentence written with red font. Is there any specific intention?

2.      Introduction:

-          statements are lack of supporting materials (no references were cited to support the claim), e.g., … (pg. 1 lines 33-34) The FO process offers high energy efficiency, low fouling propensity, high salt rejection, and low brine discharge…(says who? Any ref?)

-          (pg.1 lines 35-37)…FO process received significant attention from researchers and has been reported to have potential applications in wastewater treatment, desalination, food processing, energy production, and biomedical applications according to who?

-          Avoid using informal words like ‘not yet’.

-          Please cite necessary citation when claim is made.

-  

3.      Material and methods:

-          Figure 1, how well is the acceptance that 30ËšC is considered as hot?

4.      Result and Discussion:

-          Figure 5, significant bands were not labelled.

-          Again, claims were not support by references.

-          Table 3: Regeneration method, hot air 39-50 ËšC not stated in the methodology.

The manuscript presents a satisfactory quality for Membranes. However, the current state of the article required language editing or proofreading to improve the quality of the paper before it can be published in this respective journal.

Author Response

Rebuttal to Comments of Reviewers

(Our answers are written in italics)

Changes made in the text of the revised manuscript are marked in "BLUE".

Manuscript ID: membranes-1993034

Title: Feasibility of poly (vinyl alcohol)/ poly (diallyldimethylammonium chloride) polymeric network hydrogel as draw solute for forward osmosis process

Firstly, we would like to extend our gratitude to all the esteemed reviewers for spending their valuable time reviewing our original work and providing some constructive comments. We have incorporated all the comments in our revised manuscript.

Reviewer #1:

  1. Abstract:
  • Forward osmosis (FO) is one of the emerging technologies of what technologies?

Response: Thank you for the valuable comments. As per the reviewer's suggestions, the sentence has been re-written as:

"Forward osmosis (FO) has been identified as an emerging technology for the concentration and crystallization of aqueous solution at low temperatures. However, the application of the FO process has been limited due to the unavailability of a suitable draw solute (refer line no. 10-12)".

  • Why draw solute is important and how it can significantly affect the commercial implementation? What kind of process? Please be specific.

ResponseThank you for the valuable comments. As per the reviewer's suggestions, highlight the draw solute's importance in the forward osmosis process. The sentences have been re-written:

"An ideal draw solute should be able to generate high osmotic pressure and must be easily re-generated with less reverse solute flux (RSF). Recently, hydrogels have attracted attention as a draw solution due to their high capacity to absorb water and low RSF” (refer to lines 12-15).

  • contains a sentence written with red font. Is there any specific intention?

 Response: Thank you for the valuable comments; the mentioned sentences' colour has been changed from red to black.

  1. Introduction:
  • statements are lack of supporting materials (no references were cited to support the claim), e.g., … (pg. 1 lines 33-34) The FO process offers high energy efficiency, low fouling propensity, high salt rejection, and low brine discharge…(says who? Any ref?)

Response: Thank you for the valuable comments. The appropriate citation(s) added to support the given statement (refer to lines no. 40-46).

As pointed out by the reviewer, compared to conventional concentration techniques, the FO process is expected to be energy-efficient provided the draw solution (DS) regeneration is achieved at a low-cost, i.e. either by using waste heat resources, thermal and electrical stimulus (that requires less energy than thermal evaporation method).

  • (pg.1 lines 35-37)…FO process received significant attention from researchers and has been reported to have potential applications in wastewater treatment, desalination, food processing, energy production, and biomedical applications according to who?

Response: Thank you for the valuable suggestion, the appropriate citation(s) were added to support the given statement (refer to lines 46-49).

  • Avoid using informal words like 'not yet'.

Response: Thank you for the valuable suggestion. The negative construction words such as "not yet" have been avoided and replaced in this and in all future manuscripts of the author (s).

  • Please cite necessary citation when claim is made.

Responses: Thank you for the valuable suggestions. The citation has been revised and appropriately performed. 

  1. Material and methods:
  • Figure 1, how well is the acceptance that 30ËšC is considered as hot?

Response: Thank you for the valuable suggestions. We agree with the reviewer's observation; this study was performed at 22 ± 0.5 oC. Thus for drying the hydrogel, a temperature of 30 ± 1.5 oC was used. To maintain the uniformity of the given process condition, the drying was performed in a hot-air oven by maintaining 30 ± 1.5 oC for 8 h. (refer to lines 128-129)

  1. Result and Discussion:
  • Figure 5, significant bands were not labelled.

Response: Thank you for the valuable suggestions. The significant bands in Figure 5 (refer to Figure 5, page no. 09) have been appropriately labelled.

  • Again, claims were not support by references.

Response: Thank you for the valuable suggestions. All claims have been revised and properly cited based on reviewers' suggestions.

  • Table 3: Regeneration method, hot air 39-50 ËšC not stated in the methodology.

Response: Thank you for the valuable comment. Based on the reviewer's suggestions, the regeneration method has been introduced in the text (refer to line no. 220).

The manuscript presents a satisfactory quality for Membranes. However, the current state of the article required language editing or proofreading to improve the quality of the paper before it can be published in this respective journal.

Reviewer 2 Report

The manuscript written by Bardhan et. al presented a kind of polymer hydrogel network that can be used as the draw solute for forward osmosis processes. The authors prepared the hydrogels by crosslinking reactions between PVA, PolyDADMAC, and Glyoxal. Hydrogels with different compositions exhibited different swelling capacities. The FO experiment was conducted by a batch set-up, by wrapping up the membrane into a near-cylindrical structure. The draw solutions containing hydrogels were used inside the cylinder and the active layer was facing the draw solution. The primary feed solution is NaCl at 5 g/L. However, there is no leak-tight test demonstrating that the FO membrane is free of defects, and the NaCl rejection is not measured. The osmotic pressure of the hydrogel systems is not systematically measured, though the FO flux seems reasonable. The hydrogel networks are recovered by hot air, but the acquirement of the absorbed water was not mentioned, and this process seems to involve phase change of water, which may not be economically feasible. In the end, the authors proposed a potential process application of such hydrogel material, but that process seems not working/not energy-efficient/not economically feasible. There are also no process simulations/calculations to back up the analysis provided by the authors. This manuscript should be revised intensively, and multiple additional experiments (especially the osmotic pressures of hydrogels with different concentrations/compositions) are needed before publishing at Membranes.   

Detailed comments: 

  1. 1. The authors seem to have some misunderstandings about the concentration polarizations regarding FO membranes. FO membranes usually have 2 different CPs, namely external CP and internal CP. The ICP takes place at the support layer of the membrane, which is generally considered more severe compared to the ECP, because the ICP cannot be mitigated by adjusting external conditions, such as the increase of Reynolds number, etc.   

  1. (a) In the introduction part, line 40, the author only mentioned the ECP 

  1. (b) The reason why AL-DS is better specifically for a NaCl-feed/hydrogel draw system is that hydrogel polymer has much smaller diffusivity than NaCl. If the membrane is arranged in AL-FS the ICP effect of hydrogel will be severe. If the membrane is arranged in AL-DS the ICP will happen on the NaCl side, which will have less severe ICP since NaCl has larger diffusivity than hydrogel. 

  2.  
  1. 2. In Figure 1 the authors claimed that the polyDADMAC is added on a volume basis, but in Table 1 polyDADMAC amounts are based on wt%. What is the density of polyDADMAC? Please correct.

  2.  
  1. 3. According to Figure 3c, it is doubtful that this hand-made device is leak tight. Therefore, the FO flux may not be accurate. Perform a test with the same amount of NaCl solution as hydrogel, with the same concentration as the feed solution, and observe if there is any flux.

  2.  
  1. 4. Please also report the NaCl rejection values of the membranes for the FO tests, including the time effect and the polymer composition effect.

  2.  
  1. 5. The draw agent is recovered by hot air. Is the hot air only act as a heating method, or does it evaporate the water from the hydrogel? Can you recover the water by simply heating the hydrogel to 50 C with a water bath? If the hot air is used to evaporate the water, then the whole process is pointless --- membrane processes are developed to avoid phase change, and if the water cannot be recovered directly without any phase changes, the process is not energy efficient (even worse than direct evaporation of pure water from salt water!) Therefore, Table 3 is not a fair comparison regarding the water recovery percentage since all the other researchers use non-phase change, direct, low-energy consumption methods to recover water.

  2.  
  1. 6. Figure 6 is very confusing as the data is not matching with Table 1. Under the same composition the authors give 2 distinct values for 4:3:1.5. Please check your data and perform the measurement correctly! Also, if you would like to describe any trends, please use at least 3 experimental points. Only 2 points are not enough. Plus, in Figure 6, the compositions are not changing consecutively, so you cannot connect them by lines! Using a bar chart. If you would like to show the trend for different compositions, please plot them in separate figures, with only 1 variable changing and the others keeping constant.

  2.  

  1. 7. There are no osmotic pressures reported for different concentrations of hydrogels and different compositions of hydrogels. The reviewer believes that these data are the most important data for the publication of this paper.

  2.  
  1. 8. The hydrogel seems to be useless in the systems that the authors have proposed in Figure 9. The whole process is not backed up by any process simulations/calculations. Despite the calculation, the advantage of such a system is: “Here, due to the osmotic pressure gradient, the solvent moved from liquid food (tea) to 345 hydrogel to RO brine resulting in simultaneous dilution and concentration of RO brine 346 and liquid food while restricting reverse solute flux (RSF).” This may not sound correct. It seems that the author's original thought was to limit the RSF of NaCl since the hydrogel has a much lower RSF. But this wish may not be accomplished. (1) With the hydrogel room in the middle, the NaCl can still diffuse into the hydrogel room. The mass transfer resistance of the hydrogel room is much lower than a membrane, so the NaCl will still be able to diffuse into the tea solution. Plus, the RSF of hydrogel will pollute the tea solution twice! (2) If there are no hydrogel, just 2 layers of FO membranes, you may get better flux as the NaCl brine can supply an osmotic pressure much higher than the hydrogel, and meanwhile, may have smaller RSF of NaCl compared with only 1 layer of FO membranes. The reviewer personally believes that this part should be removed from the paper since the hydrogel has no use in such a process

Author Response

Rebuttal to Comments of Reviewers

(Our answers are written in italics)

Changes made in the text of the revised manuscript are marked in "BLUE".

Manuscript ID: membranes-1993034

Title: Feasibility of poly (vinyl alcohol)/ poly (diallyldimethylammonium chloride) polymeric network hydrogel as draw solute for forward osmosis process

Firstly, we would like to extend our gratitude to all the esteemed reviewers for spending their valuable time reviewing our original work and providing some constructive comments. We have incorporated all the comments in our revised manuscript.

Reviewer #3:

The manuscript written by Bardhan et. al presented a kind of polymer hydrogel network that can be used as the draw solute for forward osmosis processes. The authors prepared the hydrogels by crosslinking reactions between PVA, PolyDADMAC, and Glyoxal. Hydrogels with different compositions exhibited different swelling capacities. The FO experiment was conducted by a batch set-up, by wrapping up the membrane into a near-cylindrical structure. The draw solutions containing hydrogels were used inside the cylinder and the active layer was facing the draw solution. The primary feed solution is NaCl at 5 g/L. However, there is no leak-tight test demonstrating that the FO membrane is free of defects, and the NaCl rejection is not measured. The osmotic pressure of the hydrogel systems is not systematically measured, though the FO flux seems reasonable. The hydrogel networks are recovered by hot air, but the acquirement of the absorbed water was not mentioned, and this process seems to involve phase change of water, which may not be economically feasible. In the end, the authors proposed a potential process application of such hydrogel material, but that process seems not working/not energy-efficient/not economically feasible. There are also no process simulations/calculations to back up the analysis provided by the authors. This manuscript should be revised intensively, and multiple additional experiments (especially the osmotic pressures of hydrogels with different concentrations/compositions) are needed before publishing at Membranes.

Detailed comments:

  1. The authors seem to have some misunderstandings about the concentration polarizations regarding FO membranes. FO membranes usually have 2 different CPs, namely external CP and internal CP. The ICP takes place at the support layer of the membrane, which is generally considered more severe compared to the ECP, because the ICP cannot be mitigated by adjusting external conditions, such as the increase of Reynolds number, etc.

(a) In the introduction part, line 40, the author only mentioned the ECP.

Response: Thank you for the valuable comments. As per reviewers suggestions, the given sentences has been modified (refer line no. 52).

(b) The reason why AL-DS is better specifically for a NaCl-feed/hydrogel draw system is that hydrogel polymer has much smaller diffusivity than NaCl. If the membrane is arranged in AL-FS the ICP effect of hydrogel will be severe. If the membrane is arranged in AL-DS the ICP will happen on the NaCl side, which will have less severe ICP since NaCl has larger diffusivity than hydrogel.

Response: Thank you for the valuable comments. As per reviewer's suggestions, the given sentences has modified and rewritten as: “Reportedly, the due to high molecular weight and viscosity the diffusivity of hydrogel is expected to be lower than NaCl. Since ICP is the major factor contributing to the decline in the water permeation rate in FO desalination. Thus, considering the severity of the ICP -effect, the ALDS-mode of FO operation was chosen for further experimental studies”( refer line no. 164-168). 

  1. In Figure 1 the authors claimed that the polyDADMAC is added on a volume basis, but in Table 1 polyDADMAC amounts are based on wt%. What is the density of polyDADMAC? Please correct.

Response: Thank you for the valuable comments. The author(s) would like to state that the study was performed completely on a weight basis and not on volume. The author(s) have corrected the given figure (Figure 1, page 4). In this study the  PolyDADMAC (average Mw 200,000-350,000) (CAS NO. 26062-79-3 ) and the density is 1.04 g/mL at 25 °C, which is close to water (i.e. 0.997 g/mL at 25 oC).

  1. According to Figure 3c, it is doubtful that this hand-made device is leak tight. Therefore, the FO flux may not be accurate. Perform a test with the same amount of NaCl solution as hydrogel, with the same concentration as the feed solution, and observe if there is any flux.

Response: Thank you for the valuable comments. The author(s) would like to state that the leak test was performed before performing experiments. Refer to line (172-179) which states, “Figure 3 (a,b) provides a pictorial representation of the synthesized polymeric network before and after drying. The FO tests were carried out in a thimble-like FO membrane module (Figure 3 (c)). The thimble-like membrane was prepared by cutting the membrane (length: 6 cm; breadth: 3.5 cm) and glued together to form a pouch-like structure. After gluing, the active dimension of the membrane was measured accurately by excluding the glued portion (area: 17.5 cm2). Before experiments, a study was performed to detect leakage in the prepared membrane module. The module without leakage was stabilized for another 8 h using deionized distilled water (TDS: 0.445 mg L-1)”.

Additionally, the author(s) would like to highlight line no. 187-189 states, “The same batch of FO experiments was performed using high-concentration NaCl solution as DS and deionized water as FS to investigate the role of the hydrogel as a draw solute.”

Further, the change in sodium and chloride ion concentration was analyzed using Ion-chromatography. Also, using freezing-point osmometer was used to determine the osmolality of the NaCl solution. This was performed to validate the mass balance across the feed side of the system.

  1. Please also report the NaCl rejection values of the membranes for the FO tests, including the time effect and the polymer composition effect.

Response: Thank you for the valuable suggestions, the author would like to state that in the given study, commercially available membranes. The vendor(s) provided the data-sheet claimed the salt rejection as 99.4%. The same was validated by the author(s) experimentally while performing membrane characterization of virgin membrane (99.4%) and after stabilization (99.3 %).

After 12- set of FO study, the rejection studies were not performed. However, FTIR analysis (Figure 5), FESEM-analysis (Figure 4(e-f)), and leakage test were performed in order to validate the integrity of the given system.

  1. The draw agent is recovered by hot air. Is the hot air only act as a heating method, or does it evaporate the water from the hydrogel? Can you recover the water by simply heating the hydrogel to 50 C with a water bath? If the hot air is used to evaporate the water, then the whole process is pointless --- membrane processes are developed to avoid phase change, and if the water cannot be recovered directly without any phase changes, the process is not energy efficient (even worse than direct evaporation of pure water from salt water!) Therefore, Table 3 is not a fair comparison regarding the water recovery percentage since all the other researchers use non-phase change, direct, low-energy consumption methods to recover water.

Response: Thank you the valuable comments. The author(s) would like to highlight that in this study, the hot air used in this study is at 39-50 oC (which can be easily procured as waste heat from any industry). Since the source of hot air is waste heat, this procedure can be considered a feasible DS regeneration process. 

Razmjou et al. (2013) [1] stated that “water molecules to travel out of the hydrogel networks, mostly in vapor form rather than liquid form, suggesting that the regeneration mechanism is, in essence, heat conduction”. Several researchers validated the same approach the same was summarised in a recent article by Wang et al. (2020) [2].

Thereby, the author(s) would like to summarise the application of hot air (at 39-50oC) can be considered as a feasible technique for DS regeneration, provided: (i) the source of hot air ( is waste heat from an industrial source), and (ii) the water (solvent) released from gel network is condensed and used as potable water.   

Ref.

  1. Razmjou, A.; Barati, M.R.; Simon, G.P.; Suzuki, K.; Wang, H. Fast Deswelling of Nanocomposite Polymer Hydrogels via Magnetic Field-Induced Heating for Emerging FO Desalination. Environ Sci Technol 2013, 47, 6297–6305, doi:10.1021/es4005152.
  2. Wang, J.; Gao, S.; Tian, J.; Cui, F.; Shi, W. Recent Developments and Future Challenges of Hydrogels as Draw Solutes in Forward Osmosis Process. Water (Switzerland) 2020, 12, 1–20, doi:10.3390/w12030692.

  1. Figure 6 is very confusing as the data is not matching with Table 1. Under the same composition the authors give 2 distinct values for 4:3:1.5. Please check your data and perform the measurement correctly! Also, if you would like to describe any trends, please use at least 3 experimental points. Only 2 points are not enough. Plus, in Figure 6, the compositions are not changing consecutively, so you cannot connect them by lines! Using a bar chart. If you would like to show the trend for different compositions, please plot them in separate figures, with only 1 variable changing and the others keeping constant.
  2. There are no osmotic pressures reported for different concentrations of hydrogels and different compositions of hydrogels. The reviewer believes that these data are the most important data for the publication of this paper.

Response: Thank you for the valuable comments. As per the reviewers suggestion, to avoid data redundancy, Figure 6 has been removed, and the standard deviation value is added to Table 1 to attain a better understanding of the influence of hydrogel composition on swelling capacity.

The author(s) would like to state that the reported data in Table 1 are the average value of the three experiments conducted, and the result from the three experiments are mentioned below:

Hydrogel composition

Swelling capacity (g/g)

Exp01

Exp02

Exp03

Avg

Std. dev

H-3:3:1.5

13.6200

13.65

13.51

13.59

0.06

H-4:3:1.5

20.75

20.61

20.67

20.68

0.06

H-6:0:1.5

13.42

13.48

13.45

13.45

0.02

H-4:2:2

16.62

16.51

16.55

16.56

0.05

H-3:3:3

11.74

11.78

11.78

11.77

0.02

H-6:0:3

11.16

11.18

11.12

11.15

0.02

Further, the author(s) would like to highlight the importance of swelling capacity in this study. The role of hydrogels as a draw solute in the FO process has been explained in a recent review article by Wang et al. (2020) [1] (refer to lines no, 75-76), and the critical findings from the mentioned literature have been summarised as below:

  • Reportedly, similar to osmotic pressure, the hydrogel swelling pressure contributes to the equilibration of chemical potential between the inner and outer volume of swollen hydrogel [2].
  • In the case of hydrogel, when a dry polymer comes in contact with water, a certain amount of water is first absorbed by capillary and physical diffusion. Subsequently, the hydrophilic group enhances the affinity to water by hydrogen bonds and the solvation effect.
  • Due to the ionization effect, many same charges and counter-ions are produced. However, the counter-ions cannot migrate freely outside the polymeric network to maintain electrical neutrality.

This enhances osmotic pressure inside the polymeric network, thus driving more water molecules into the polymeric network.

  • However, with an expansion of the polymeric network, the elastic contraction force caused between the molecular chains limits the expansion of the polymeric network. As a result, the hydrogel achieves swelling equilibrium.
  • We agree with the reviewer that the hydrogel is not liquid and cannot be circulated through the membrane module. Thus the hydrogel (draw solute) needs to be regenerated within the membrane module itself (using warm air (refer to section 2.6) or high-concentration brine solution (Figure 8(a)).

Thus, this suggests that osmotic pressure is the true driving force for hydrogel swelling.

Additionally, the author wants to state that the FTIR, IC, and osmolality of feed solution (as de-ionized water) were performed before and after each batch of experiments to validate no significant leaching of components from polymeric network to feed solution. Similarly, the mass balance for the feed solution (NaCl) was performed to validate that the water absorbed by hydrogel is free from sodium or chloride ion (refer to lines no. 187-189).

Ref.

  1. Wang, J.; Gao, S.; Tian, J.; Cui, F.; Shi, W. Recent Developments and Future Challenges of Hydrogels as Draw Solutes in Forward Osmosis Process. Water (Switzerland) 2020, 12, 1–20, doi:10.3390/w12030692.
  2. Höhne, P.; Tauer, K. How Much Weighs the Swelling Pressure. Colloid Polym Sci 2014, 292, 2983–2992, doi:10.1007/s00396-014-3347-0.

  1. The hydrogel seems to be useless in the systems that the authors have proposed in Figure. The whole process is not backed up by any process simulations/calculations. Despite the calculation, the advantage of such a system is: “Here, due to the osmotic pressure gradient, the solvent moved from liquid food (tea) to 345 hydrogel to RO brine resulting in simultaneous dilution and concentration of RO brine 346 and liquid food while restricting reverse solute flux (RSF).” This may not sound correct. It seems that the author's original thought was to limit the RSF of NaCl since the hydrogel has a much lower RSF. But this wish may not be accomplished. (1) With the hydrogel room in the middle, the NaCl can still diffuse into the hydrogel room. The mass transfer resistance of the hydrogel room is much lower than a membrane, so the NaCl will still be able to diffuse into the tea solution. Plus, the RSF of hydrogel will pollute the tea solution twice! (2) If there are no hydrogel, just 2 layers of FO membranes, you may get better flux as the NaCl brine can supply an osmotic pressure much higher than the hydrogel, and meanwhile, may have smaller RSF of NaCl compared with only 1 layer of FO membranes. The reviewer personally believes that this part should be removed from the paper since the hydrogel has no use in such a process.

Response: Thank you for the valuable comments. The author(s) agree with the reviewers' concern regarding the potential RSF propensity and the importance of the given 3-tier membrane module (Figure 8(a)). This section proposes the prospects of using synthesized hydrogel for liquid food concentration. The membrane in this study has salt rejection ability, and hydrogel results in comparatively lower RSF. Thus it was assumed that the given module would give lower RSF compared to the scenario when high-concentration NaCl is directly used as DS.

The data and ideas reported in this study are at the preliminary stages. However, with great pleasure, the author would like to state by improving the FO performance of the synthesized hydrogel, the stated idea was successfully implemented for the concentration of liquid (tea extract). Using the given module, the author(s) were able to concentrate tea extract to 4.5 fold (with RSF of 0.107 g m-2 h-1). The detailed results and observations are summarised in the author(s) future manuscript. (Note: The figure below provides a brief overview of the study's flux performance).

The detailed discussion and evaluation results are part of the author's future manuscript. This manuscript focuses on the preliminary studies, whereas the future manuscript focuses on applying hydrogel as a draw solute for the concentration of liquid food extract.

Reviewer 3 Report

The paper entitled ''Feasibility of poly (vinyl alcohol)/ poly (diallyldimethylammonium chloride) polymeric network hydrogel as draw solute for forward osmosis process'' investigates the feasibility of a draw solution (DS) synthesized from polymeric materials for forward osmosis (FO) process. FO is an emerging membrane-based water treatment technique driven by the osmotic pressure gradient. So, DS plays a critical role in the performance of FO systems. Although the topic of the study is interesting, the manuscript content requires a major improvement. The authors poorly evaluated the FO treatment applicability. The manuscript mostly includes the results of material characterization. Further FO membrane experiments should be performed to conclude that the use of the synthesized material in FO water treatment applications is an advantage. I am not sure this material can be used in FO water treatment applications as a draw solution since FO has two liquid-circulating parts. However, this material is a gel that can not be circulated through the stream. My comments are as follows.

1.     Abstract, Lines 15-16: The authors should give evaluations based on the results rather than general statements. Please mention how this material improved the FO performance.

2.     Abstract, Lines 20-21: It needs to be explained how the authors achieved that hydrogel can be used as a draw solution. This statement needs to be proven.

3.     Introduction, Lines 56-66: The authors mentioned that hydrogels entrap a large amount of water, and ionic species induce the hydrogel to swell. The role of hydrogels in FO treatment should be given. I think it can not be used as a draw solution because it is not acting as a liquid. It just absorbs water. Also, I am not sure the absorbed water is free of ions. This should be proven as well.

4.     Figure 2, Please mention in the text how hydrogel was placed on the membrane surface.

5.     Sections 2.6 and 3.4: This section is not about life cycle assessment. Please look at the definition of LCA.

6.     Figure 7: The flux is very low. Please explain how it can be used for the scale-up of the process.

7.     Figure 8C: Images are low quality, it is difficult to evaluate the images.

8.     Section 3.5: A very limited amount of results have been given in this section to evaluate its performance in practical applications. More results should be given in this section.

9.     Add a summarizing sentence on what should be remembered after each section in results and discussion.

10.  Conclusion: This section should be written by evaluating interesting results. The format that concise text followed by bullets is recommended.

Author Response

Rebuttal to Comments of Reviewers

(Our answers are written in italics)

Changes made in the text of the revised manuscript are marked in "BLUE".

Manuscript ID: membranes-1993034

Title: Feasibility of poly (vinyl alcohol)/ poly (diallyldimethylammonium chloride) polymeric network hydrogel as draw solute for forward osmosis process

Firstly, we would like to extend our gratitude to all the esteemed reviewers for spending their valuable time reviewing our original work and providing some constructive comments. We have incorporated all the comments in our revised manuscript.

Reviewer#2

The paper entitled ''Feasibility of poly (vinyl alcohol)/ poly (diallyldimethylammonium chloride) polymeric network hydrogel as draw solute for forward osmosis process'' investigates the feasibility of a draw solution (DS) synthesised from polymeric materials for forward osmosis (FO) process. FO is an emerging membrane-based water treatment technique driven by the osmotic pressure gradient. So, DS plays a critical role in the performance of FO systems. Although the topic of the study is interesting, the manuscript content requires a major improvement. The authors poorly evaluated the FO treatment applicability. The manuscript mostly includes the results of material characterisation. Further FO membrane experiments should be performed to conclude that the use of the synthesised material in FO water treatment applications is an advantage. I am not sure this material can be used in FO water treatment applications as a draw solution since FO has two liquid-circulating parts. However, this material is a gel that cannot be circulated through the stream.

My comments are as follows.

  • Abstract, Lines 15-16: The authors should give evaluations based on the results rather than general statements. Please mention how this material improved the FO performance.

Response: Thank you for the valuable comments. The sentence has been re-written as:

"The result indicates that incorporating cationic polyelectrolyte poly (diallyldime-thylammonium chloride) into the polymeric network has increased the swelling capacity and osmotic pressure. Thereby resulting in an average water flux of PVA-polyDADMAC hydrogel (0.972 L m-2 h-1) that was 7.47 times higher than the PVA-hydrogel during 6 h FO process against 5000 mg L-1 NaCl solution (as feed solution)" (refer line no. 21-23).

  • Abstract, Lines 20-21: It needs to be explained how the authors achieved that hydrogel can be used as a draw solution. This statement needs to be proven.

Response: Thank you for the valuable comments. As per the given comments, the following are the changes made in the given manuscript:

  • The author(s) would like to draw reviewers' attention toward Section 3.5 (lines 389-414), where the prospects of application of the hydrogel (as draw solution) for liquid food concentration has been discussed in detail.

Note: The author(s) would like to state that the detailed related results have been incorporated in the author(s) future manuscript. The figure below provides a brief overview of the FO performance of hydrogel (as draw solute) used for the concentration of tea extract using the FO process.

  • Also, the result indicates that incorporating cationic polyelectrolyte poly (diallyldimethylammonium chloride) into the polymeric network has increased the swelling capacity and osmotic pressure. Thereby resulting in an average water flux of PVA-polyDADMAC hydrogel (0.97 L m-2 h-1) that was 7.47 times higher than the PVA-hydrogel during 6 h FO process against 5000 mg L-1 NaCl solution (as feed solution) (refer line no. 21-23).
  • At 50oC, the hydrogel releases nearly >70 % of the water absorbed during the FO process at room temperature. Water flux can be recovered to 86.6% of the initial flux after 12-times of hydrogel (draw solute) regeneration. Furthermore, the study suggests that incorporating cationic polyelectrolyte into the polymeric network enhances the FO performances and lowers the actual energy requirements for (draw solute) regeneration (refer to lines 25-31).

  • Introduction, Lines 56-66: The authors mentioned that hydrogels entrap a large amount of water, and ionic species induce the hydrogel to swell. The role of hydrogels in FO treatment should be given. I think it cannot be used as a draw solution because it is not acting as a liquid. It just absorbs water. Also, I am not sure the absorbed water is free of ions. This should be proven as well.

Response: Thank you for the valuable comments. The role of hydrogels as a draw solute in the FO process has been explained in a recent review article by Wang et al. (2020) [1] (refer to lines no, 75-77), and the important findings from the mentioned literature have been summarised as below:

  • Reportedly, similar to osmotic pressure, the hydrogel swelling pressure contributes to the equilibration of chemical potential between the inner and outer volume of swollen hydrogel [2].
  • In the case of hydrogel, when a dry polymer comes in contact with water, a certain amount of water is first absorbed by capillary and physical diffusion. Subsequently, the hydrophilic group enhances the affinity to water by hydrogen bonds and the solvation effect.
  • Due to the ionisation effect, many same charges and counter-ions are produced. However, the counter-ions cannot migrate freely outside the polymeric network to maintain electrical neutrality.

This enhances osmotic pressure inside the polymeric network, thus driving more water molecules into the polymeric network.

  • However, with an expansion of the polymeric network, the elastic contraction force caused between the molecular chains limits the expansion of the polymeric network. As a result, the hydrogel achieves swelling equilibrium.
  • We agree with the reviewer that the hydrogel is not liquid and cannot be circulated through the membrane module. Thus the hydrogel (draw solute) needs to be regenerated within the membrane module itself (using warm air (refer to section 2.6) or high-concentration brine solution (Figure 9(a)).

Thus, this suggests that osmotic pressure is the true driving force for hydrogel swelling.

Additionally, the author wants to state that the FTIR, IC, and osmolality of feed solution (as de-ionised water) were performed before and after each batch of experiments to validate no significant leaching of components from polymeric network to feed solution. Similarly, the mass balance for the feed solution (NaCl) was performed to validate that the water absorbed by hydrogel is free from sodium or chloride ion (refer to lines no. 187-189).

  1. Wang, J.; Gao, S.; Tian, J.; Cui, F.; Shi, W. Recent Developments and Future Challenges of Hydrogels as Draw Solutes in Forward Osmosis Process. Water (Switzerland) 2020, 12, 1–20, doi:10.3390/w12030692.
  2. Höhne, P.; Tauer, K. How Much Weighs the Swelling Pressure. Colloid Polym Sci 2014, 292, 2983–2992, doi:10.1007/s00396-014-3347-0.

  • Figure 2, Please mention in the text how hydrogel was placed on the membrane surface.

Response: Thank you for the valuable comments. Based on the reviewer's suggestions, the methodology has now been introduced in the text (refer to lines 154-157).

  • Sections 2.6 and 3.4: This section is not about life cycle assessment. Please look at the definition of LCA.

Response: Thank you for the valuable comments. Based on the reviewer's suggestion, the title of the given section has been changed to "Hydrogel stability test" (refer to section 2.6 and section 3.4).

  1. Figure 7: The flux is very low. Please explain how it can be used for the scale-up of the process.

Response: Thank you for the valuable comments. In context to the given comments, the author(s) would like to draw the reviewer's kind attention toward:

  • Table 3 (Hydrogel as draw solute for the FO process). Compared to the reported hydrogel, though the permeate flux is low (1.81 L m-2 h-1, in first 1 h against 2500 mg L-1 NaCl solution as feed solution), the water recovery rate was considerably higher (> 70%; at 39-50oC).
  • Figure 8 states that water flux can be recovered to 86.6% of the initial flux after 12-time hydrogel (draw solute regeneration).
  • The author(s) would like to state that using the current hydrogel, the freshly brewed tea extract can be potentially concentrated up to 4.5 fold (RSF, 0.107 g m-2 h-1). Further, the author(s) would like to mention that the author(s) performed further experiments related to this topic and will be submitting the manuscript soon.

Thereby, taking context to the above-mentions points, the scale-up would be a practical approach, considering the authors concluding remark to improve the FO performance of the synthesised flux in future studies.

  1. Figure 8C: Images are low quality, it is difficult to evaluate the images.

Response: Thank you for the valuable comments. Based on the reviewer's suggestions, the low-quality digital microscopic image has been replaced with a high-resolution image (refer to figure 8(c), Page no. 13).

  1. Section 3.5: A very limited amount of results have been given in this section to evaluate its performance in practical applications. More results should be given in this section.

Response: Thank you for the valuable comments. This section provides an overview of the prospects of synthesised hydrogel as DS in liquid food concentration applications. The detailed discussion and evaluation results are part of the author's future manuscript. This manuscript focuses on the preliminary studies, whereas the future manuscript focuses on applying hydrogel as a draw solute for the concentration of liquid food extract.

  1. Add a summarising sentence on what should be remembered after each section in results and discussion.

Response: Thank you for the valuable comment. Based on the reviewer's valuable comments, the authors have added summarising statements after each section in the results and discussion.

  1. Conclusion: This section should be written by evaluating interesting results. The format that concise text followed by bullets is recommended.

Response: Thank you for the valuable comment. The conclusion section has been edited in bullet format (refer to lines no. 416-428).

Round 2

Reviewer 2 Report

Most of the comments have been addressed. However, the reviewer still does not agree with the reply to comments 7 and 8.

For comment 7, the osmotic pressure may not be directly measured, but can you use equilibrium measurement (e.g, use a NaCl draw solution and test the osmotic pressure of NaCl when there is no flux between NaCl and the hydrogel)

For comment 8, the reviewer still believes this process will not significantly reduce RSF. If you use 2 layers of FO membrane (instead of 1 layer of FO membrane ) and NaCl as the draw solute, you will still have a smaller RSF. Adding the hydrogels in the middle between 2 FO membranes will not help.

Author Response

Rebuttal to Comments of Reviewers

(Our answers are written in italics)

Changes made in the text of the revised manuscript are marked in "BLUE".

Manuscript ID: membranes-1993034

Title: Feasibility of poly (vinyl alcohol)/ poly (diallyldimethylammonium chloride) polymeric network hydrogel as draw solute for forward osmosis process

Firstly, we would like to extend our gratitude to all the esteemed reviewers for spending their valuable time reviewing our original work and providing some constructive comments. We have incorporated all the comments in our revised manuscript.

Reviewer #1:

Most of the comments have been addressed. However, the reviewer still does not agree with the reply to comments 7 and 8.

  • For comment 7, the osmotic pressure may not be directly measured, but can you use equilibrium measurement (e.g, use a NaCl draw solution and test the osmotic pressure of NaCl when there is no flux between NaCl and the hydrogel).

Response: Thank you for the valuable comments. We used reviewer recommended approach to estimate the osmotic pressure of hydrogel, the experimentally, we observed that while maintaining 6.305 bar NaCl FS, the water flux was observed to be near zero. However, this pressure cannot be taken as the actual osmotic pressure of hydrogel due to the ICP and ECP effect. To incorporate same in the manuscript, the following text is included, i.e., lines 381-383, which state,"The osmotic pressure for the synthesized hydrogel was approximately 6.305 bar. The osmotic pressure of the synthesized hydrogel (DS) can be estimated by its potential of drawing solvent from FS (NaCl solution, 0.5 g L-1 to 10 g L-1)."

Based on the flux data from FO experiments, the hydrogel's osmotic pressure was estimated to be 6.305 bar (which is equivalent to 8000 mg L-1 NaCl at 25oC temperature).

For comment 8, the reviewer still believes this process will not significantly reduce RSF. If you use 2 layers of FO membrane (instead of 1 layer of FO membrane) and NaCl as the draw solute, you will still have a smaller RSF. Adding the hydrogels in the middle between 2 FO membranes will not help.

Response: Thank you for the valuable comments. With great pleasure, the authors would like to state that the experiments were performed using the given hydrogel and achieved considerably superior FO performance. The detailed experimental results are reported in our future manuscript. However, the following are some highlights from the observed results:

  • While the author(s) have used FO set-up (Figure R1(a)), the SRSF was reported to be 0.107 g L-1, and flux performance is given in Figure R1(b).

Figure R1. (a)  FO test set-up, and (b) FO performance when 10g L -1 Tea as FS and high-concentration NaCl solution as 'regeneration fluid.'

  • While much higher flux performance (4.5 times higher) was achieved when the FO module, as shown in Figure 8 (a,b), was used. The detailed explanations are part of the author(s) future manuscript.

Also, while using NaCl as a "hydrogel (DS) regenerative solution", due to the low mass transfer coefficient across the NaCl solution-UF membrane-hydrogel layer, the NaCl flux (i.e., RSF) across these layers is expected to be very low. As per our initial experimental study with a three-layer module having 38.4 g L-1 (osmotic pressure, 30.27 bar) NaCl – hydrogel- distilled water in FO experiment, till 6 hr of the batch experiment, we observed 0.05 g m-2 h-1 of NaCl flux (RSF), which is very minimal compared to other cases 2.81 g m-2 h-1 (i.e., NaCl directly as DS without hydrogel). The author(s) believe that the NaCl diffusion coefficient in the hydrogel is exceptionally low compared to the NaCl diffusion coefficient in water, enabling low RSF.

Reviewer 3 Report

I have carefully reviewed the R1 version of the manuscript. The authors have considerably provided a detailed and thorough response to my comments by addressing my major concerns and have improved the manuscript accordingly. I have some minor comments at this stage to further improve the quality of the manuscript.

1.      Please mention in the text that the hydrogel as a draw solution is not liquid and can not be circulated in the system, and explain why it is considered a draw solution.

2.      I am not sure the hydrogel can stay on the membrane surface during FO operation. The driving force due to the flow velocity of the solution will remove it from the surface. Please mention how to overcome this challenge during the FO operation.

Author Response

Rebuttal to Comments of Reviewers

(Our answers are written in italics)

Changes made in the text of the revised manuscript are marked in "BLUE".

Manuscript ID: membranes-1993034

Title: Feasibility of poly (vinyl alcohol)/ poly (diallyldimethylammonium chloride) polymeric network hydrogel as draw solute for forward osmosis process

Firstly, we would like to extend our gratitude to all the esteemed reviewers for spending their valuable time reviewing our original work and providing some constructive comments. We have incorporated all the comments in our revised manuscript.

Reviewer #3:

I have carefully reviewed the R1 version of the manuscript. The authors have considerably provided a detailed and thorough response to my comments by addressing my major concerns and have improved the manuscript accordingly. I have some minor comments at this stage to further improve the quality of the manuscript.

  1. Please mention in the text that the hydrogel as a draw solution is not liquid and cannot be circulated in the system, and explain why it is considered a draw solution.

Response: Thank you for the valuable suggestions. As per the reviewer's suggestion, the text has been modified (refer to lines 76-77) with appropriate citations, and the related explanation is incorporated into the text (refer to lines 60-95).

  1. I am not sure the hydrogel can stay on the membrane surface during FO operation. The driving force due to the flow velocity of the solution will remove it from the surface. Please mention how to overcome this challenge during the FO operation.

Response: Thank you for the valuable suggestions. The author(s) would like to state that in a semi-swollen state, the hydrogels are placed in the membrane module using a flat-surfaced spatula (refer 157-161). Also, in the 3-tier membrane module (Figure 8 (a,b)), using a flat-surfaced spatula, the hydrogel was introduced to the middle tier (height, 1.5 cm). While introducing to the middle tier, the semi-swollen hydrogel was tightly packed to ensure their uniform contact with the membrane.

Since the hydrogel in the middle was semi-swollen, tightly packed to ensure contact with the membrane surface. Due to the viscous nature of the hydrogel and the minimal height of the middle tier (1.5 cm), the hydrogels were intact in their position.

The detailed mechanisms, experimental details, and explanation are part of the author(s) future manuscript.
